# Cornering the universal shape of fluctuations

Benoit Estienne [1✉], Jean-Marie Stéphan [2✉] & William Witczak-Krempa [3,4,5✉]

Understanding the fluctuations of observables is one of the main goals in science, be it theoretical or experimental, quantum or classical. We investigate such fluctuations in a subregion of the full system, focusing on geometries with sharp corners. We report that the angle dependence is super-universal: up to a numerical prefactor, this function does not depend on anything, provided the system under study is uniform, isotropic, and correlations do not decay too slowly. The prefactor contains important physical information: we show in particular that it gives access to the long-wavelength limit of the structure factor. We exemplify our findings with fractional quantum Hall states, topological insulators, scale invariant quantum critical theories, and metals. We suggest experimental tests, and antici- pate that our findings can be generalized to other spatial dimensions or geometries. In addition, we highlight the similarities of the fluctuation shape dependence with findings relating to quantum entanglement measures.

[1] Sorbonne Université, CNRS, Laboratoire de Physique Théorique et Hautes Énergies, LPTHE, F-75005 Paris, France. [2] Univ Lyon, CNRS, Université Claude Bernard Lyon 1, UMR5208, Institut Camille Jordan, F-69622 Villeurbanne, France. [3] Département de Physique, Université de Montréal, Montréal, QC H3C 3J7, Canada. [4] Centre de Recherches Mathématiques, Université de Montréal, P.O. Box 6128, Centre-ville Station, Montréal, QC H3C 3J7, Canada. [5] Regroupement Québécois sur les Matériaux de Pointe (RQMP), Montreal, QC, Canada. ✉email: estienne@lpthe.jussieu.fr; stephan@math.univ-lyon1.fr; w.witczak-krempa@umontreal.ca

In quantum mechanics, measurements on identically prepared systems of an observable $\mathcal{O}$ will generally yield different outcomes. This is a consequence of the fact that the state of the system is in a quantum superposition of states having well-defined values of $\mathcal{O}$. The spread of the outcomes, ignoring experimental errors, can be quantified by the variance, or uncertainty squared in the quantum language, $(\Delta\mathcal{O})^2 = \langle(\mathcal{O} - \langle\mathcal{O}\rangle)^2\rangle$. Heuristically, we say that $\Delta\mathcal{O}$ measures the fluctuations of $\mathcal{O}$. Similar fluctuations also occur in classical many-body systems, where the statistical description leads to fluctuations of observables. In numerous experiments, like scanning tunneling microscopy, one only measures a small subregion of a sample. In that case, a natural question arises: What are the fluctuations of a given observable in a subregion $A$? This refinement introduces additional information: the shape of the subregion. It thus seems that one is left with a huge amount of possibilities corresponding to different quantum or classical states, observables, and shapes, and thus little hope to find unifying principles[1–3]. In this work, we show that there exists a large, and experimentally relevant, set of states and observables that share the same universal shape dependence for their fluctuations.

Let us consider a local scalar observable, written in the continuum as $\rho(\mathbf{r})$. It could be the number of bacteria per unit area, the charge density, the energy density, the local magnetization, etc. The fluctuations of $\rho$ within a subregion $A$ are described by $\Delta\mathcal{O}_A$, where $\mathcal{O}_A = \int_A d\mathbf{r}\,\rho(\mathbf{r})$ is the integrated density in the subregion:

$$(\Delta\mathcal{O}_A)^2 = \langle\mathcal{O}_A^2\rangle - \langle\mathcal{O}_A\rangle^2 = \int_A d\mathbf{r}\int_A d\mathbf{r}'\langle\rho(\mathbf{r})\rho(\mathbf{r}')\rangle_c \quad (1)$$

with the connected correlation function $\langle\rho(\mathbf{r})\rho(\mathbf{r}')\rangle_c = \langle\rho(\mathbf{r})\rho(\mathbf{r}')\rangle - \langle\rho(\mathbf{r})\rangle\langle\rho(\mathbf{r}')\rangle$. The expectation value is taken either with respect to a classical distribution, or a quantum density matrix. We will now focus on a uniform and isotropic systems, for which the above correlation function only depends on the distance separating the two positions $\langle\rho(\mathbf{r})\rho(\mathbf{r}')\rangle_c = f(|\mathbf{r} - \mathbf{r}'|)$, yielding

$$(\Delta\mathcal{O}_A)^2 = \int_A d\mathbf{r}\int_A d\mathbf{r}'\,f(|\mathbf{r} - \mathbf{r}'|)\,. \quad (2)$$

The function $f$ can be very different depending on the system and choice of observable and is generally not known for realistic models. From general principles, fluctuations of most physical systems behave for large regions $A$ as[4]

$$(\Delta\mathcal{O}_A)^2 = \alpha|A| + \beta|\partial A| - b_A + \cdots\,. \quad (3)$$

The first term is a standard volume law, scaling with the size of $A$, while the second term is an area law scaling with the size of its boundary $\partial A$, $b_A$ the dominant subleading term, and the ellipses denote weaker terms, in particular those that vanish in the thermodynamic limit. The prefactors $\alpha$ and $\beta$ do not depend on the shape of region $A$, and they can be explicitly computed in terms of the correlation function $f$ (see Supplementary Note 1). The subleading term $b_A$ is more interesting: it carries the non-trivial shape dependence of the fluctuations and probes the large-scale properties of the system. In particular, if $A$ has sharp corners, each one contributes to $b_A$. These corner contributions are encoded in a function $b(\theta)$, $\theta$ being the corner opening angle. The case of a simple planar corner in two dimensions is illustrated in Fig. 1.

In this work, we report that the angle-dependence $b(\theta)$ of the fluctuations is in fact completely independent of the observable and of the system considered, up to a numerical prefactor.

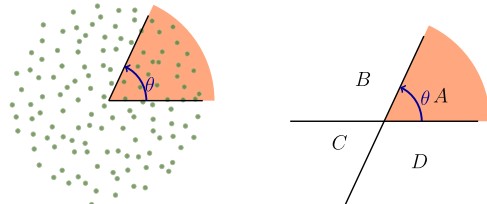

**Fig. 1 Corner geometry.** Left: Subregion $A$ is highlighted in red. The electron distribution is a typical Monte Carlo sample obtained from the topological fractional quantum Hall ground state at filling $\nu = 1/3$. Right: The regions that are used in the substraction procedure for canceling out the boundary law, and isolating the corner contribution.

Namely

$$b(\theta) = -(1 + (\pi - \theta)\cot\theta)\int_0^\infty \frac{r^3}{2}f(r)\,dr \quad (4)$$

as long as the system is translation invariant and isotropic, and the correlation function $f$ decays sufficiently fast at large $r$, as discussed below. We emphasize that the aforementioned assumptions have considerable generality: they hold for a wide class of classical and quantum systems, at zero or finite temperature. A typical example would be that of a liquid, where in addition to translational invariance and isotropy, $f$ decays exponentially fast. We test our super-universal result with various systems. We first consider fractional quantum Hall systems, where even the prefactor of the corner term is universal and proportional to the Hall conductivity. This example can also be interpreted as a classical (liquid) particle system with 2d Coulomb repulsion via the plasma analogy. We then examine quantum critical scale-invariant theories, for which the corner function diverges logarithmically: the prefactor is also universal in that case, but has a different physical origin. It is proportional to the longitudinal conductivity. We next investigate the case of metals, which breaks our assumptions and shows different behavior. Finally, we present large-scale quantum Monte Carlo results for the corner contribution to the entanglement entropy of an FQH ground state, which provides new support for the connection between quantum entanglement and fluctuations[5–7].

## Results

Strikingly, the simple angle dependence factorizes and is independent of the correlation function $f$. Before we provide the derivation of this result and present several non-trivial tests, it is worthwhile to pause and examine the angular function in Eq. (4), $u(\theta) = 1 + (\pi - \theta)\cot\theta$, which we call the corner fluctuation function. It is plotted in Fig. 2 (left). Due to the appearance of the cotangent, $\cot\theta = \frac{\cos\theta}{\sin\theta}$, it diverges as $1/\theta$ when the angle approaches zero. The increase at small angles is natural given that the region is becoming thinner, which leads to stronger long-range fluctuations. In the opposite limit of $\theta \approx \pi$, it vanishes quadratically as $(\theta - \pi)^2$.

The prefactor of the corner fluctuation function, that is the radial integral in Eq. (4), is also meaningful and holds interesting physical information. This coefficient can be measured experimentally, as it is directly related to the long-wavelength limit of the static structure factor, which can be accessed via elastic scattering experiments, for example. We shall treat various examples below. The reader may remark that the integral is free from large-scale divergence provided $f$ decays faster than $1/r^4$. When the decay is precisely $1/r^4$, one obtains a logarithmic divergence with the size of region $A$, as we shall explain when we treat scale-invariant quantum critical systems. However, in

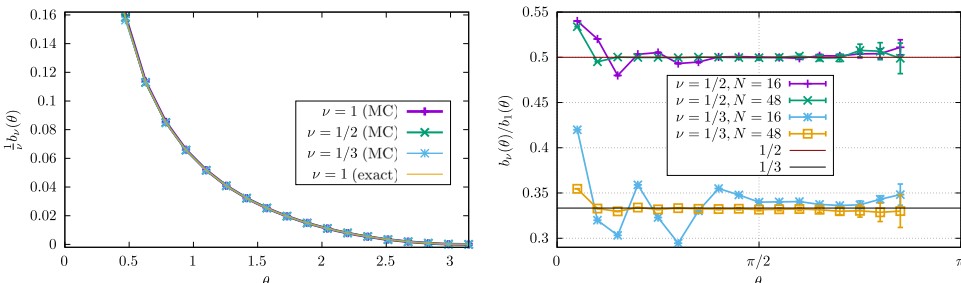

**Fig. 2 Charge fluctuations in fractional quantum Hall states.** Monte Carlo (MC) extraction of the corner term $b_\nu$ for several filling fractions $\nu$. Left: $\nu^{-1}b_\nu(\theta)$ for $N = 48$ particles and filling fractions $\nu = 1, 1/2, 1/3$. The collapse onto a single curve is nearly perfect. Right: Illustration of the finite-size effects for $\nu = 1/2, 1/3$, by plotting $b_\nu(\theta)/b_1(\theta)$ for an increasing number of particles. The curves become constant for large $N$. The slight discrepancy for small $\theta$ is a finite size effect: for such small angles, there are very few particles in $A$, unless $N$ is extremely large. Note also the increase of the error bars when $\theta$ becomes close to $\pi$. This is an artifact of the fact that $b_\nu$ vanishes in that limit: while absolute error bars (of statistical origin, as described in the "Methods" section) are still very small, relative errors blow up.

certain situations, the decay is even slower, and we will obtain a new scaling.

The corner fluctuation function has previously appeared in several contexts, for example in renormalization studies of Wilson loops in gauge theories[8,9], in the study of entanglement entropy[5,10,11], in bipartite fluctuations of non-interacting Dirac systems[7] and the integer quantum Hall effect[12], and in the study of disorder operators in two-dimensional conformal field theories[13,14]. Our findings illuminate its physical origin in a large class of classical and quantum systems and explain why it has appeared in these seemingly unrelated contexts.

In order to evaluate the corner contribution $b(\theta)$, we consider for region $A$ a single corner of opening angle $\theta$, as illustrated in Fig. 1 (left). First, we must isolate the subleading corner contribution from the volume and boundary terms. The volume law is easy to take care of. Its origin can be traced back to the fluctuations of the total integrated density $\mathcal{O} = \int_V d\mathbf{r}\rho(\mathbf{r})$. Indeed, unless $\mathcal{O}$ is conserved, its variance is extensive with the total system size $V$, with a density of fluctuations (see e.g. ref. [4]) $\alpha = (\Delta\mathcal{O})^2/V = \int d\mathbf{r}\langle\rho(\mathbf{r})\rho(\mathbf{0})\rangle_c$. Straightforward manipulations lead to

$$(\Delta\mathcal{O}_A)^2 = \alpha|A| - \int_A d\mathbf{r}\int_{A^c} d\mathbf{r}'\, f(|\mathbf{r} - \mathbf{r}'|) \qquad (5)$$

where $A^c$ denotes the complement of region $A$. This takes care of the volume term, the second term in the r.h.s. being governed by an area law provided $f$ decays sufficiently fast to zero. In particular, if $\mathcal{O}$ does not fluctuate, the volume contribution vanishes. This is for example the case for ground states of local Hamiltonians that respect the symmetry corresponding to $\mathcal{O}$. In the appropriate temperature regime, the fluctuations of $\mathcal{O}_A$ then mainly occur due to the motion of the local charge in the immediate vicinity of the boundary, leading to a boundary law. It is important to emphasize that boundary law can also dominate in a variety of contexts beyond the low-temperature limit, such as in certain excited states. Care must be taken at finite temperature in the canonical ensemble. In this case, $\alpha$ will vanish but the function $f$ will not decay to zero, resulting in a separate volume term. This can be easily seen at infinite temperature, where correlations do not quite vanish due to the constraint on particle number in the whole system. In contrast, $\alpha$ is the only possible volume law contribution in the grand-canonical ensemble.

We next have to cancel out the boundary term in the second term in Eq. (5), which we call $\Theta_A$. To do so we consider a subtraction scheme based on a four-corner geometry, as illustrated in Fig. 1 (right). Because the subregions $A, B, C$ and $D$ have an infinite boundary, the quantities $\Theta_A, \Theta_B, \ldots$ are also infinite. But these boundary contributions cancel out in the following linear

combination, leaving only the subleading angle-dependent correction:
$b(\theta) = \frac{1}{2}(\Theta_{AB} + \Theta_{AD} - \Theta_A - \Theta_C) = -\int_B d\mathbf{r}\int_D d\mathbf{r}' f(|\mathbf{r} - \mathbf{r}'|)$.
This integral is evaluated in Supplementary Note 2, where an alternative derivation, not relying on a substraction procedure, is also presented. Both methods lead to the universal corner fluctuation function (4). As long as the correlation function $f(r)$ decays fast enough at long distances, the radial integral in Eq. (4) is convergent. This is guaranteed for example, but not exclusively, for gapped states. This integral can generally be measured experimentally as it is directly related to the long-wavelength limit of the static structure factor $S(\mathbf{k}) = \int e^{i\mathbf{k}\cdot\mathbf{r}}\langle\rho(\mathbf{r})\rho(\mathbf{0})\rangle_c\, d\mathbf{r}$ associated to the observable $\mathcal{O}$:

$$S(\mathbf{k} \to 0) = S(0) - \pi k^2 \int_0^\infty \frac{r^3}{2}f(r)\, dr \qquad (6)$$

Note also that $S(0) = \alpha$ gives the coefficient of the volume term. This is natural, as bipartite fluctuations over large regions can probe the long-wavelength limit of the static structure factor[15].

We now test the super-universal shape dependence in a variety of systems, starting with quantum Hall states, and topological insulators.

**Fractional quantum Hall liquids.** Two-dimensional classical liquids and gapped quantum phases provide a broad and natural class of systems for which our results directly apply. In addition to being homogeneous and isotropic, their correlation function $f(r)$ typically decays exponentially. An interesting example is provided by fractional quantum Hall states. These states are topological phases of electrons moving in two dimensions at low temperatures under the influence of a strong transverse magnetic field. They host anyon quasiparticles that are neither fermions nor bosons and support gapless chiral edge modes. We will study their charge fluctuations. It is known[16–19] that for incompressible phases, the static structure factor takes the following form at small wavevectors: $S(\mathbf{k} \to 0) = l_B^2 k^2 \langle\rho\rangle/2$, where $l_B$ is the magnetic length and $\langle\rho\rangle = \nu/2\pi l_B^2$ is the electron density. The filling fraction $\nu$ gives the number of electrons per quantum of magnetic flux. Using this result (also called a sum rule) allows us to write the full corner term:

$$b_\nu(\theta) = \frac{\nu}{4\pi^2}(1 + (\pi - \theta)\cot\theta) = \frac{|\sigma_{xy}|}{2\pi}(1 + (\pi - \theta)\cot\theta) . \qquad (7)$$

In the last equality, we have related the filling fraction to the Hall conductivity in natural units, $e = \hbar = 1$. For the integer quantum Hall effect at $\nu = 1$, this was previously derived[12]. But in fact,

Eq. (7) is valid for general incompressible interacting ground-states, including abelian and non-abelian topological states.

Let us illustrate this general result with the example of the Laughlin state, written in first quantization as

$$\Psi(z_1, \dots, z_N) = \prod_{1 \le i < j \le N} (z_i - z_j)^{1/\nu} e^{-\frac{1}{4}\sum_{i=1}^N |z_i|^2} , \qquad (8)$$

for integer values of $1/\nu$. The coordinate $z_j = x_j + iy_j$ of the $j$th electron is expressed as a complex number, and lengths are measured in terms of the magnetic length. This state is a seminal example of a fractional quantum Hall fluid. For $1/\nu \ge 2$ it has intrinsic topological order giving rise to abelian anyon quasi-particles, while $\nu = 1$ describes the non-interacting integer quantum Hall effect. For large $N$, the particles lie in a droplet of radius $\sqrt{2N/\nu}$. In the bulk, the particle density is uniform, with exponentially decaying correlations. The conducting edge excitations are described by a chiral conformal field theory. Since quantum Hall states are gapped, fluctuation of particle numbers in region $A$ are expected to obey an area law. For the integer quantum Hall effect, this is rigorously established[20], while for the fractional case this area law has been confirmed numerically[21]. Furthermore, in the non-interacting case, the connected two-point function is known exactly so the integral (4) can be readily computed, confirming the result (7) (see Supplementary Note 3). However, such an elementary derivation is not viable in the interacting case, since the two-point function $f(r)$ is not known, and besides the ones we are using, only a few sum rules are known (e.g. refs. [22–27]).

We check the corner function (4) using Monte Carlo simulations in order to sample the many-body wavefunction (8). We work with filling fractions $\nu = 1/3$ and $1/2$, which correspond to topologically ordered ground states for fermions, and bosons, respectively. We compute the particle variance in a given subregion, with the only complication being that simulation time can become large to have sufficient precision on the variance. Data shown in this section are typically averaged over several billion samples. Another complication comes from the edge of the droplet, which hosts gapless chiral modes. Fortunately, the contribution from these modes is known exactly and has been shown[12] to decouple from the corner contribution, so we can easily substract it (see the "Methods" section). For a sufficiently large particle number, the corner contribution $b_\nu(\theta)$ is given by (4), as shown in Fig. 2, confirming our arguments to high precision.

Our discussion of fluctuations in topological states has so far been limited to systems that break time-reversal. However, similar results will hold for non-chiral states. For instance, let us consider a model wavefunction for a fractional topological insulator[28], which consists of two decoupled FQH states, formed by spin up and down electrons with Hall conductivities of opposite signs. To be concrete, this could be realized by putting each spin species in a Laughlin state with a filling fraction of 1/3. Due to the decoupling of the up and down spins, the charge variance is simply twice that of a single Laughlin state. Indeed, the charge variance is invariant under time reversal. This is why it is really the absolute value of the Hall conductivity that appears in (7). We thus find that the first equality in Eq. (7) for the universal shape dependence holds, but with the replacement $\nu \to 2\nu$. The second equality should also be modified for this family of non-chiral topological insulators since $\sigma_{xy}$ vanishes: the prefactor is now proportional to the spin Hall conductivity.

We close this section by mentioning that we have verified that the super-universal shape dependence also holds for an infinite family of excited, and thermal quantum Hall states. Let us first consider excited states at unit filling by entirely occupying only

the $n$th Landau level, with $n > 0$, and leaving all other levels empty (see Supplementary Note 3). We find that those states obey the angle dependence Eq. (4), but with $\nu$ in the prefactor replaced by $2n + 1$. We thus see that for excited states, the prefactor is no longer simply given by the filling. It is expected that the charge fluctuations increase with the energy of the excited state. It would be interesting to understand the prefactor for other uniform excited states.

We now consider the integer quantum Hall effect at a finite temperature $T$ and chemical potential $\mu$. The corresponding correlation function $f(r)$ still decays exponentially at large distances, which ensures that the charge fluctuations obey the super-universal shape dependence (4). By evaluating the prefactor at small temperatures, we find that it remains unchanged up to corrections that are exponentially small in the ratio of the cyclotron energy to twice the thermal energy, $\hbar\omega_c/(2k_B T)$. As the temperature increases towards the cyclotron scale and beyond, the prefactor varies in a non-trivial way, which can be determined; we provide additional information in Supplementary Note 3.

**Scale-invariant quantum critical theories**. After having studied gapped topological phases, we now turn to a large family of gapless systems: quantum critical phases and phase transitions. We shall focus on systems with emergent Lorentz and scale invariance, which in the majority of cases combine to an even larger conformal symmetry. The gapless Dirac cones of graphene or the quantum critical transition between an insulator and superfluid at integer filling constitute key examples[29]. Furthermore, the symmetries of the overarching conformal field theory impose the large distance behavior for the correlation function of a conserved global charge to be $f(r) = -C_J/r^4$ [30]. $C_J$ is a positive constant that gives the universal ground state longitudinal conductivity in natural units, $\sigma = \pi^2 C_J/2$, of the associated conserved current. For such systems, the variance of a conserved charge obeys a strict area law (Supplementary Note 4). To find the subleading correction, we substitute this $f$ into Eq. (4) and get

$$b(\theta) = \frac{\sigma}{\pi^2}(1 + (\pi - \theta)\cot\theta) \log(|\partial A|/\delta) . \qquad (9)$$

The result grows logarithmically with the perimeter of $A$; we have introduced a short-distance cutoff, $\delta$. This scaling with the perimeter is in contrast to the constant $b(\theta)$ for gapped systems. We note that the prefactor of the logarithm is entirely universal since it is not polluted by microscopic details (here represented by the cutoff $\delta$). We stress that the expression (9) holds for any conformal field theory, irrespective of how strongly correlated it is. Interestingly, whereas the universal Hall conductivity appeared in the corner fluctuations of quantum Hall groundstates Eq. (7), the above equation features the universal longitudinal conductivity, illustrating that universality can arise from different origins. In the specific case of non-interacting Dirac fermions, Eq. (9) was previously obtained[7].

Let us consider a different observable that is present in all CFTs, namely the energy density. The conformal symmetry constrains the two-point function to be $f(r) = 2C_T/(3r^6)$ [30], where $C_T$ is a positive coefficient that depends on the theory. As the $f$ function decays sufficiently rapidly at large distances, using Eq. (4) we obtain a corner term with the super-universal angle dependence, with a prefactor that is constant with respect to the size of region $A$, in contrast to what was found above for a global charge (9). Another difference with Eq. (9) is that the prefactor is no longer universal as it depends on microscopic information (the short-distance cutoff). In fact, we can consider infinitely many other observables, in which case the correlation function $f$ scales as $1/r^{2\Delta}$, where $\Delta$ is the scaling dimension of the

observable. As long as $\Delta \geq 2$, we obtain the super universal fluctuation function (9), with a prefactor that depends on the microscopic details unless $\Delta = 2$. This later case shows the importance of global symmetries in the study of bipartite fluctuations. In conformal quantum critical theories, there exists a small number of observables with $\Delta < 2$, and the corresponding correlation functions decay more slowly at large distances. Interestingly, a slow decay also occurs for charge fluctuations in metals. As we discuss next, this leads to a qualitatively distinct geometrical dependence for the fluctuations.

**Metals.** As the last example, we study fluctuations in metals. These have more mobile excitations at low energies compared to the quantum critical theories described above. For example, a two-dimensional metal with a circular Fermi surface has an entire Fermi line worth of gapless points in momentum space, whereas Dirac semimetals only have a finite number of discrete gap-closing points. As such, it is not surprising that metals have stronger charge fluctuations than scale-invariant critical systems. For regular metals, called Fermi-liquids, the dominant contribution to the charge fluctuations has a logarithmic enhancement compared to the boundary law, $|\partial A| \ln |\partial A|$, with a prefactor which is known analytically[31,32]. This enhancement is related to the fact that $f(r)$ decays slower than for charge fluctuations of CFTs at large separations, namely as $1/r^3$. This has further important consequences as we now discuss.

For subsystem $A$, it is convenient to take a circular sector with radius $L$ and opening angle $\theta$. As stated before, the dominant term is a logarithmically enhanced boundary law. We identified the first subleading correction, which is proportional to $L$. It is given by

$$b(\theta) = L\, b_{FL}(\theta) \qquad (10)$$

where an explicit formula for $b_{FL}$ is given in Supplementary Note 5. It is different from the super-universal corner function discussed above. We stress that this term depends on the full geometry of $A$, as well as the shape of the Fermi surface. As such it becomes a full geometric term, rather than a simple corner term. Such behavior is related to the long-range decay of the correlation function, which blurs the notion of locality necessary to define a corner contribution. In particular, there is no reason for it to vanish at $\theta = \pi$, and be symmetric under $\theta \to 2\pi - \theta$, as before. We show in Supplementary Note 5 that this is true only up to an additive contribution, which is affine in $\theta$. The function $b_{FL}$ does nevertheless contain interesting information. For example at small angles, $b_{FL}$ diverges logarithmically instead of the previous $1/\theta$ scaling, which further illustrates the difference with the super-universal corner function studied in this paper.

It would be interesting to investigate how much of this picture changes for non-Fermi-liquids, such as the fermionic half-filled Landau level[33], which can have a different decay of charge correlations.

**Fluctuations and entanglement.** One motivation of the present study is the connection between bipartite fluctuations and quantum entanglement that is emerging from various directions[4,7,15,31]. Indeed, the corner fluctuation function behaves almost identically as the entanglement entropy of various systems including scale-invariant quantum critical points, non-interacting Dirac fermions, integer quantum Hall groundstates, and super-symmetric gauge theories dual to certain string theories[5,6,34,35]. The entanglement entropy[36] captures the amount of uncertainty for any measurement spatially localized to a subregion of the system and is dominated by quantum entanglement at low temperature. However, due to the difficulty in studying the

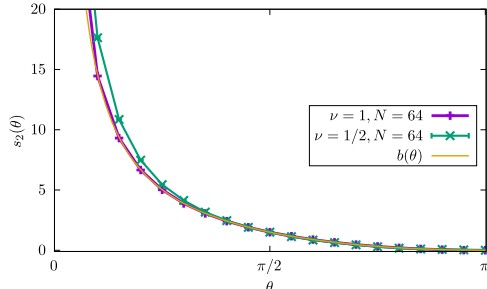

**Fig. 3 Comparing the shape dependence of fluctuations and entanglement.** Corner contribution $s_2(\theta)$ to the second entanglement Rényi entropy for quantum Hall states as a function of the subregion's corner angle, obtained using Monte Carlo simulations at fillings $\nu = 1$ and $1/2$ with $N = 64$ particles. The solid orange line corresponds to the super-universal fluctuation function, Eq. (4). All curves have been normalized to the unit second derivative at $\theta = \pi$. As in the case of fluctuations, the boundary law contribution has been removed. The error bars are of statistical origin, as described in the "Methods" section.

entanglement entropy in the many-body setting, the universality of the fluctuation-entanglement connection remains unclear. For example, the corner dependence of the entanglement entropy in gapped interacting systems has never been computed. Here, we use large-scale Monte Carlo simulations to compute the second Rényi entropy $S_2$ for the gapped FQH state at filling $\nu = 1/2$ discussed above. Monte Carlo simulations of entanglement entropies are computationally much more expensive, but can nevertheless be carried out using the swap method[37]. The resulting shape dependence of both quantities is shown in Fig. 3: close agreement is observed for a large range of angles. Note that both quantities have been normalized so that they behave as $(\theta - \pi)^2$ as $\theta \to \pi$. Moreover, they show the same asymptotic behavior in $1/\theta$ for small angles. It would be desirable to study the entanglement-fluctuation connection in other systems and to shed light on its origin.

We note that a distinct connection between the entanglement entropy and local observables exists for certain quantum critical points possessing a purely spatial conformal symmetry[38]. There, the entanglement entropy of a subregion is given by the free energy (not fluctuations) of another classical two-dimensional theory defined on the subregion and its complement[39]. The corresponding entanglement corner function[38], although different, behaves similarly to the super-universal fluctuation function described in the present paper.

## Discussion

We have seen how the shape of fluctuations of an observable $\rho(\mathbf{r})$ in a subregion with corners becomes super universal, i.e. it takes the same form for a very large class of unrelated systems. In fact, the systems could be classical or quantum. We have theoretically tested our result using quantum Hall states, both fractional and integer, topological insulators, and scale-invariant quantum critical theories. It would be interesting to further test this super-universality in the laboratory. On the classical front, one could study the number fluctuations of colloidal particles at a two-dimensional interface (such as air/water). It should be possible to use microscopy to determine the shape dependence of the particle variance for subregions with varying corner angles. On the quantum front, a natural testbed would be ultracold atomic gases loaded in an optical lattice. Using various shapes of subregions one would be able to probe the atom number variance in phases like the Mott insulator or at the superfluid-to-insulator (conformal) quantum critical point[40,41].

Our analysis was mainly in two dimensions, but such super-universality is bound to occur in higher dimensions as well. We give one concrete example in three dimensions: take subregion $A$ to be a solid cone of opening angle $\theta$, i.e. a 2d corner rotated about its axis of symmetry. For simplicity, let us consider the variance of a conserved charge in scale-invariant quantum critical theories described by a CFT, such as a three-dimensional Dirac semimetal. In these CFTs, the symmetry enforces the connected correlation function to scale as $1/r^6$. Mapping the fluctuation calculation to the one for the entanglement entropy[42] of a special model (Supplementary Material), we find that all such quantum critical theories will receive a correction that scales as $\frac{\cos^2(\theta/2)}{\sin(\theta/2)}(\log|\partial A|)^2$, with the prefactor being given by the universal groundstate conductivity of the system. This result is thus very similar to what we have obtained in two dimensions, Eq. (9). This universal cone fluctuation function agrees with the specific example of Dirac fermions in three dimensions[7], and holds for arbitrary CFTs. We conjecture that it will arise in the fluctuations of many other systems. Interestingly, the cone function is the same one (up to a prefactor) that characterizes the entanglement entropy of conical subregions in general CFTs[43]. It would be of interest to further investigate the universality of this result and to also examine other geometries, such as trihedral corners appearing in polyhedra-like cubes or tetrahedra.

Finally, we have seen that the fluctuations offer a unique window into the intricate realm of quantum entanglement, but with the advantage of being much simpler to obtain both theoretically and experimentally. Our work raises the important question: Why do these distinct quantities, computed in a large variety of systems, obey nearly the same shape dependence? Insights regarding this question will help us understand the super-universal structure that emerges in many-body systems.

## Methods

We provide more details on the numerical extraction of the corner term in the fractional quantum Hall effect. The main complication stems from the fact that the pair correlation function is not quite a translational invariant for large but still finite $N$. It is in the bulk of the droplet, but there are non-trivial power-law correlations at the edge. This edge behavior is well known to be described by a chiral CFT[44], and results in an extra contribution to the charge fluctuations[12]. Fortunately, this contribution decouples from the corner term, since correlations decay exponentially fast in the bulk.

In our geometry with opening angle $\theta$, the charge fluctuations are expected to scale as

$$(\Delta N_A^\theta)^2 = \alpha'\sqrt{N} + \frac{\nu}{2\pi^2}\log\left(\sqrt{N}\sin\frac{\theta}{2}\right) - b_\nu(\theta) + \text{cst} + \ldots \qquad (11)$$

for the Laughlin state. Neither the area law prefactor $\alpha'$ nor the last constant depend on $\theta$. The logarithmic term is typical for a one-dimensional CFT. We note that the interpretation of the factor $\nu$ is slightly different for this term since it is the Luttinger parameter of the underlying free boson CFT. It is straightforward to extract the corner term using the above result. Provided $N$ is large enough, we have

$$b_\nu(\theta) = (\Delta N_A^\theta)^2 - (\Delta N_A^\pi)^2 - \frac{\nu}{2\pi^2}\log\left(\sin\frac{\theta}{2}\right), \qquad (12)$$

where $b_\nu(\theta)$, is given by Eq. (4) with the coefficient obtained in Eq. (7). $(\Delta N_A^\theta)^2$ can be evaluated numerically using standard Markov chain Monte Carlo techniques, and from this, we reconstruct the r.h.s. of the previous equation, which is shown in Fig. 2.

A similar procedure can be implemented to extract the corner contribution to the entanglement entropy, (see e.g. ref. [12]). Numerical evaluation of the second Rényi entropy can be performed using the swap method, as explained in ref. [37], but requires more computer effort. This is mainly due to the fact that one cannot directly access the entropy $S_2$, but rather $e^{-S_2}$, which is a small number, so typically requires greater statistics. All error bars were obtained by running several long independent simulations and computing the standard deviation between the results of each simulation. The data shown in Fig. 3 corresponds to over $10^{10}$ samples—requiring several CPU-years—with error bars not visible to the eye. Note that we also checked the stability of the curves with respect to particle number $N$.

## Data availability

The data that support the findings of this study are available from the corresponding author upon reasonable request.

## Code availability

All numerical codes in this paper are available upon reasonable request to the authors.

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

## Acknowledgements
This project was funded by a grant from Fondation Courtois, a Discovery Grant from NSERC, a Canada Research Chair, and a "Etablissement de nouveaux chercheurs et de nouvelles chercheuses universitaires" grant from FRQNT. B.E. was supported by Grant No. ANR-17-CE30-0013-01. J.-M.S. was supported by IDEX Lyon project ToRe (Contract No. ANR-16-IDEX-0005). We are grateful to Semyon Klevtsov for discussions about sum rules in fractional quantum Hall states, and to Y.-C. Wang, M. Cheng, and Z.Y. Meng for sharing their draft with us.

## Author contributions
B.E., J.-M.S., and W.W.-K. were responsible for the original idea for the project. All authors participated equally in the analytical calculations and wrote the manuscript together. J.-M.S. performed the Monte Carlo simulations.

## Competing interests
The authors declare no competing interests.
