## [Peer Review File · Nature Communications]

REVIEWERS' COMMENTS

Reviewer #1 (Remarks to the Author):

The authors have made many interesting changes to the manuscript and clarified my previous confusions, meanwhile adding a significant amount of material that strengthens the results reported, most notably in checking the results for fractional spin-hall states.

I think these results are general and should be of interest to a wide audience. I therefore support the publication of the manuscript in its current form in nature communications.

Reviewer #2 (Remarks to the Author):

Dear editor,

The authors study fluctuations of observables in a very general setting. Interestingly, the results they obtain are extremely general, or 'super universal'. Independent of the type of system, the fluctuations of a subregion with a sharp corner, only depend on the opening angle and a physical pre-factor (related to the structure factor), provided rather mild conditions are obeyed.

The importance of fluctuations in general, and the broad range of systems for which the authors obtain their results, make this a very interesting paper indeed. It is well written, and provides enough detail to follow and understand the results obtained. Given that this paper was refereed before, I do not have many technical issues to comment on. I tried to judge to what extent the authors incorporated the comments/recommendations by the previous referees, though I was hampered by the fact these reports were not available fully, only the parts that the authors incorporated in their reply. In any case, I deem that the changes made to the manuscript clearly improved the manuscript, both content wise as well as its readability.

The authors consider both critical systems, and (now) the connection with entanglement entropy. Given this, I think the paper would benefit from a short discussion on the connection of the results obtained with the results by Fradkin and Moore (PRL 97, 050404 (2006)), which in turn are based on the work by Cardy and Peschel (Nucl. Phys. B300, 377 (1988)).

In conclusion, I found it very interesting and enjoyable to read this paper, containing rather general results on the important topic of fluctuations. I recommend this paper for publication in nature communications.

Reply to Referee Reports

Referee Report #1

We thank the Referee for his/her his careful reading of the new version, and support for publication.

Referee Report #2

We are delighted to read the positive remarks about our work, and are grateful for the recommendation of publication.

We thank the Referee for his insightful remark regarding the connection to important works by Fradkin, Moore and Cardy, Peschel. We have added a paragraph to explain the connection, at the level appropriate for the journal, and we have added the corresponding references. Please see the new paragraph in red on page 7.